# Protocol for the development of a procedure guide on Laparoscopic Cholecystectomy: Beyond bile duct injury prevention

Camilo Ramírez-Giraldo [1,2]*, Daniela Álvarez-León[2], Alejandro Karduss-López[2], on behalf of Bogotá Task Force Collaboration Group¶

1 Department of Surgery, Hospital Universitario Mayor - Méderi, Bogotá, Colombia, 2 Escuela de Medicina y ciencias de la Salud, Universidad del Rosario, Bogotá, Colombia

¶ Bogotá Task Force Collaboration Group in the S2 appendix.
* ramirezgiraldocamilo@gmail.com

## Abstract

### Background

Laparoscopic cholecystectomy is one of the most frequently performed operations worldwide and the standard treatment for benign gallbladder disease. Although several "safe cholecystectomy" initiatives aim to prevent bile duct injury, current guidelines do not comprehensively address the intraoperative technical conduct of the procedure. Key elements—including exposure, dissection techniques, technical sequencing, and intraoperative decision-making—remain inconsistently defined. This protocol describes the development of an evidence-based, globally applicable clinical procedure guideline intended to standardize technical performance, enhance patient safety, and support surgical training and quality improvement.

### Methods

The guideline will be developed following AGREE II standards to ensure methodological rigor, transparency, and stakeholder involvement. The Guideline Development Group comprises coordinators, a steering committee, and multidisciplinary experts from surgical societies and academic institutions. Clinical questions will be generated using the PICO framework and refined through a Delphi consensus process following the ACCORD guideline. For each question, systematic literature searches will be conducted in accordance with PRISMA standards. When feasible, evidence will be synthesized using pairwise or network meta-analysis (PRISMA-NMA, PRISMA-Search); when quantitative synthesis is not possible, findings will be summarized narratively using SWiM guidance. Study selection, data extraction, and risk-of-bias assessment will be performed independently using validated tools (ROB 2.0, ROBINS-I, AMSTAR-2), with search management in Rayyan®. Recommendations will be developed using the GRADE approach, with evidence profiles created

**Data availability statement:** No datasets were generated or analyzed for the present protocol manuscript. Upon completion of the study and publication of the final guideline, deidentified study materials will be made publicly available. These materials may include anonymized Delphi response data, extracted evidence tables, and GRADE evidence profile and Summary of Findings tables, as applicable. Data and supporting materials will be shared through the Open Science Framework and/or as supplementary files accompanying the final publication.

**Funding:** The author(s) received no specific funding for this work.

**Competing interests:** The authors have declared that no competing interests exist.

in GRADEpro GDT. A second Delphi process will be conducted to reach consensus on each recommendation, applying predefined thresholds for participation and agreement.

## Discussion

This guideline seeks to fill a critical gap in the technical standardization of laparoscopic cholecystectomy by providing a comprehensive, evidence-based framework that extends beyond bile duct injury prevention. Existing guidelines lack methodological rigor and fail to address key intraoperative elements such as exposure, dissection strategies, and decision-making. Through systematic evidence synthesis and consensus processes, this project aims to harmonize global surgical practice and establish a benchmark for training, safety, and quality improvement.

## Registration

This protocol was prospectively registered in the Open Science Framework on December 18, 2025 (https://doi.org/10.17605/OSF.IO/78QSE).

## Background

Laparoscopic cholecystectomy is the gold standard for the surgical management of benign gallbladder disease and represents one of the most frequently performed laparoscopic procedures worldwide. Its widespread adoption reflects both its proven efficacy and the global evolution toward minimally invasive surgical techniques. The scale of this procedure is remarkable: in England, 1,234,319 gallbladder surgeries were performed between 2000 and 2019, averaging 61,716 per year [1]. In the United States, estimates indicate between 650,000 and 700,000 cholecystectomies are conducted annually [2]. Similarly, high volumes have been reported in Asian and European contexts—over 70,000 cases per year in Korea [3], and 47,912 procedures in Sweden between 2007 and 2010 [4]. In Latin America, data from Colombia show 192,002 cholecystectomies performed between 2012 and 2016 under the contributory health insurance system [5], highlighting the global relevance of this operation across diverse healthcare systems.

Despite being frequently performed, laparoscopic cholecystectomy is not without risk. One of its most serious complications, bile duct injury, occurs in approximately 0.4–0.6% of cases and can lead to significant morbidity [6]. In response, several "safe cholecystectomy" programs and international guidelines have been developed to prevent such injuries [7–9]. However, while these initiatives primarily focus on avoiding bile duct damage, other crucial technical and intraoperative aspects—such as dissection strategies, exposure techniques, and intraoperative decision-making—remain insufficiently addressed in current guidelines. These factors are essential not only for preventing biliary injury but also for reducing other complications, including hemorrhage, bile leakage, and surgical site infections.

In this context, the objective of this guideline is to develop a globally applicable, evidence-based framework that provides structured recommendations for key technical and intraoperative aspects of laparoscopic cholecystectomy, including operative exposure, dissection strategies, use of adjunctive techniques when appropriate, and decision-making in difficult scenarios, with the aim of reducing complications, enhancing patient safety, and promoting consistent, high-quality surgical performance across institutions.

## Methods

A clinical procedure guideline for laparoscopic cholecystectomy will be developed with the primary objective of standardizing surgical practice, improving patient safety, and promoting evidence-based decision-making in the operative management of benign gallbladder disease. This guideline will provide structured, transparent, and scientifically grounded recommendations for surgeons, researchers, and educators in the field of general and minimally invasive surgery worldwide. Recommendations will not address special populations such as pediatric or pregnant patients, as these groups fall outside the intended scope of this guideline.

The development process will follow the methodological standards established by the AGREE II (Appraisal of Guidelines for Research and Evaluation) [10,11] instrument, ensuring clarity, methodological rigor, stakeholder involvement, applicability, and editorial independence.

The launch of the guideline will take place during the 10th CIMED Research Meeting, on November 14, 2025, in Bogotá, Colombia. During this event, the objectives, methodological framework, and initial phases of the guideline development will be formally presented to the academic and surgical community, marking the official start of its implementation.

This initiative represents a collaborative effort among clinical experts, academic institutions, and research groups committed to improving the quality and consistency of surgical care through the creation of a rigorously developed, evidence-based procedural guideline.

The protocol for this guideline has been registered in the Open Science Framework (https://doi.org/10.17605/OSF.IO/78QSE).

This protocol was assessed by the corresponding institutional ethics committee and was approved through an review process (approval code: CEISH-2026051; Minutes No. 040–2026). The study constitutes a low-risk methodological guideline-development project involving voluntary participation of expert panelists, with no patient enrollment, clinical intervention, or collection of sensitive personal data. Delphi responses were collected and analyzed in deidentified form.

Fig 1 summarizes the overall development process, from establishment of the guideline development group and question formulation to evidence synthesis, recommendation development, consensus, approval, and dissemination.

### Stage 1: Establishment development group composition

The Guideline Development Group is composed of coordinators, a steering committee, and invited participants. The coordinators and steering committee include surgeons from the Hospital Universitario Mayor–Méderi, Universidad del Rosario, and the Colombian Association of Surgery, supported by a team providing statistical and methodological expertise.

The coordinators and steering committee are responsible for: establishing objectives and timelines, including protocol registration and publication; formulating the initial list of clinical questions; conducting structured consensus processes to refine and validate these questions and recommendations; reporting findings in a formal statement and guideline document; disseminating the guideline through open-access publication, scientific meetings, and online professional networks.

Participants include members of the Colombian Association of Surgery and external experts invited by the steering committee and coordinators. The target panel size of approximately 60 participants was selected to ensure broad representation across institutions, healthcare settings, and geographic regions, while maintaining feasibility and adequate participation throughout the Delphi process. External experts are invited by the steering committee based on predefined criteria, including recognized expertise in laparoscopic cholecystectomy and/or hepatobiliary

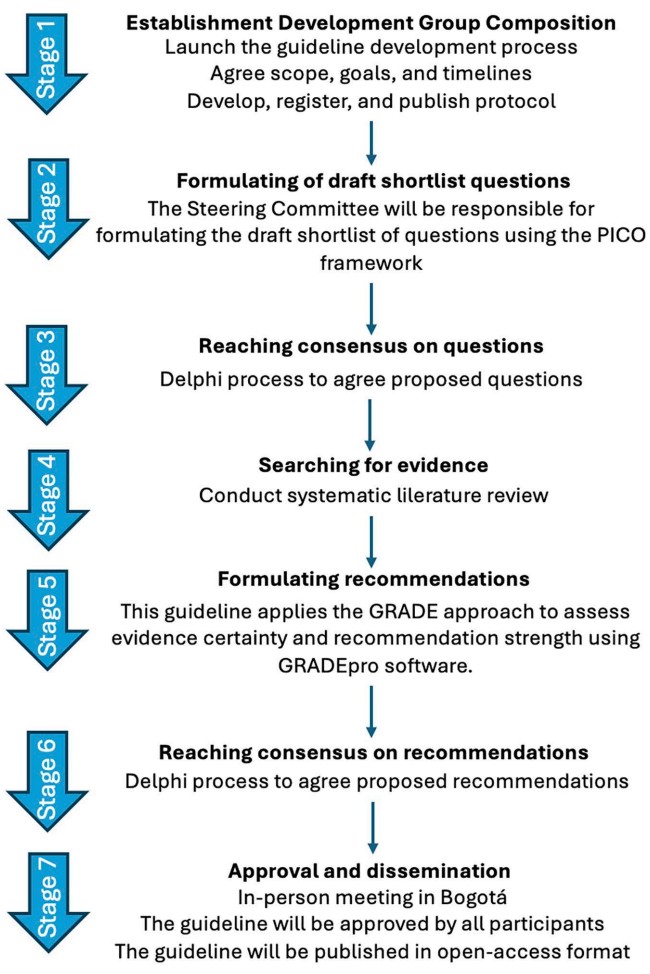

**Stage 1**
**Establishment Development Group Composition**
Launch the guideline development process
Agree scope, goals, and timelines
Develop, register, and publish protocol

**Stage 2**
**Formulating of draft shortlist questions**
The Steering Committee will be responsible for formulating the draft shortlist of questions using the PICO framework

**Stage 3**
**Reaching consensus on questions**
Delphi process to agree proposed questions

**Stage 4**
**Searching for evidence**
Conduct systematic literature review

**Stage 5**
**Formulating recommendations**
This guideline applies the GRADE approach to assess evidence certainty and recommendation strength using GRADEpro software.

**Stage 6**
**Reaching consensus on recommendations**
Delphi process to agree proposed recommendations

**Stage 7**
**Approval and dissemination**
In-person meeting in Bogotá
The guideline will be approved by all participants
The guideline will be published in open-access format

**Fig 1. Overview of the Bogotá procedure guideline development process for laparoscopic cholecystectomy.**

surgery, academic and research experience, years of clinical practice, and involvement in surgical education or quality improvement initiatives. Efforts are made to include participants from different countries and healthcare systems to enhance the global relevance and applicability of the guideline. This composition was considered appropriate because hepatobiliary and experienced laparoscopic surgeons are directly involved in the clinical and technical aspects addressed by this procedure guideline. Methodological support will be provided by team members with experience in evidence synthesis, critical appraisal, and guideline development methodology. Additionally, anesthesiologists were invited to contribute to questions specifically related to perioperative analgesia and anesthetic management during the procedure. No formal public or private stakeholder groups were involved in the development of this protocol.

All coordinators, steering committee members, and invited participants will be asked to disclose potential financial and non-financial conflicts of interest before participating in the guideline development process. These disclosures will be reviewed and documented by the coordinators and steering committee and will be considered throughout the different phases of the project. When deemed relevant, participants with potential conflicts of interest may be restricted from contributing to or voting on specific questions, evidence summaries, or recommendations. Conflict of interest

declarations and their management will be monitored throughout the development process and reported in the final guideline.

## Stage 2. Formulating of draft shortlist questions

The Steering Committee will formulate the draft shortlist of key clinical questions, which will serve as the foundation of this guideline. These questions will focus on predefined intraoperative domains of laparoscopic cholecystectomy, including access and setup, operative exposure, dissection techniques, use of adjunctive intraoperative strategies, and bailout approaches in difficult situations. The guideline will not include questions related to the underlying pathology leading to surgery, preoperative preparation, or postoperative management, as these areas lie beyond its defined scope. However, selected anesthesia-related issues may be considered when they directly affect intraoperative safety or procedural execution, such as pneumoperitoneum pressure, positioning-related physiological effects, or analgesic interventions performed during the operation. Broader aspects of preoperative anesthetic assessment and postoperative pain management pathways will remain outside the scope of this guideline.

Each question will be developed using the PICO framework (Population, Intervention, Comparator, Outcome) to ensure precision, transparency, and methodological rigor. This approach facilitates the identification of clinically relevant issues and guides the subsequent systematic literature searches.

By engaging surgeons, researchers, and surgical educators, the process aims to generate clinically meaningful and evidence-based questions that reflect real-world challenges in laparoscopic cholecystectomy. The preliminary list of candidate clinical questions formulated by the Steering Committee is provided in the supplementary material (S1 Appendix).

## Stage 3: Reaching consensus on questions

Fig 2 illustrates the Delphi process used to refine and select key clinical questions, including anonymous voting, feedback review, and predefined consensus thresholds.

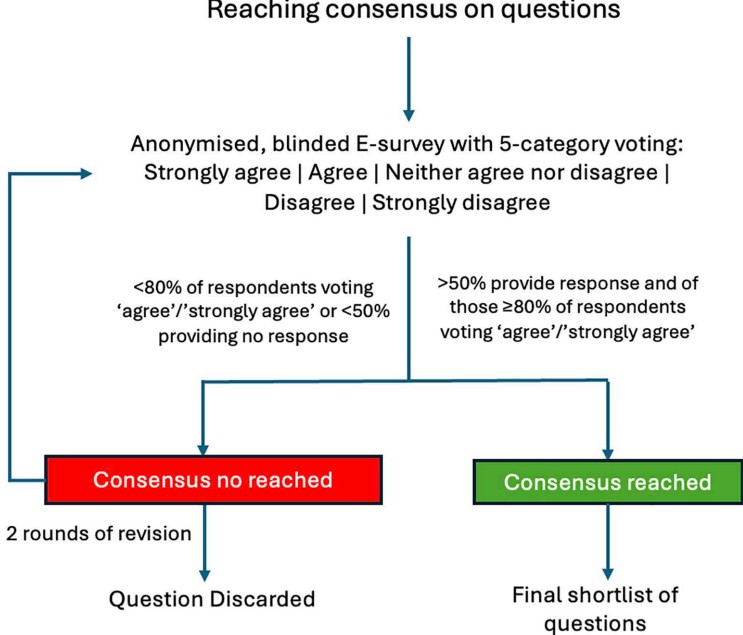

**Fig 2. Delphi consensus process for selecting key clinical questions.**

A Delphi methodology, reported in accordance with the ACCORD guideline [12], will be applied to achieve expert consensus on each clinical question. The preliminary list, developed by the Steering Committee and coordinators, will be submitted to the participants for validation using a blinded electronic voting platform (MicrosoftForms).

The order of questions will be randomized to ensure independent evaluation. Agreement will be rated using a five-point Likert scale ("Strongly agree," "Agree," "Neither agree nor disagree," "Disagree," "Strongly disagree"), with "Neither agree nor disagree" included in the denominator. Panelists will also classify each question as "critical", "important", or "not important".

Experts may provide free-text comments, propose additional questions, or recommend deletions. Two Delphi rounds will be conducted, integrating feedback and descriptive statistics after each round, analyzed by CR-G and AK-L. Approval rates and reasons for exclusion will be reported. Items not reaching consensus after a Delphi round will be reviewed by the coordinators and steering committee together with the accompanying free-text comments. Questions may be modified when comments suggest ambiguity, overlap, wording problems, or the need to better align the item with the scope of the guideline. Questions may be dropped if they remain below the predefined consensus threshold and are judged to have limited relevance or insufficient support from the panel. Questions considered clinically relevant but still unresolved after two rounds may be deferred to future updates or documented as unresolved items. Questions predominantly rated as 'not important' will not be prioritized for full evidence synthesis unless the steering committee identifies a compelling clinical or methodological reason for their retention.

The consensus threshold will be defined as at least 50% panel participation and ≥80% agreement ("Agree" or "Strongly agree"). The ≥50% participation threshold was selected to balance methodological rigor with the practical challenges of maintaining engagement across multiple Delphi rounds in a large and geographically diverse expert panel. Consensus will not be determined by participation alone but will also require a high agreement threshold of ≥80% among respondents. A participation rate of exactly 50% will be considered acceptable for that round. Reminder notifications will be sent during each Delphi round to optimize participation and reduce attrition. If participation falls below 50%, the round will remain open for an additional reminder period before finalizing the results. Attrition between Delphi rounds will be documented, and response rates will be reported for each round. When feasible, respondent characteristics, including geographic region, practice setting, and professional background, will be summarized descriptively to assess the representativeness of the participating panel. Questions failing to reach consensus or lacking sufficient participation after this process will be excluded from the final guideline.

Descriptive statistics—including response proportions, agreement medians, and interquartile ranges—will be used to assess consensus stability between rounds.

The participants will include approximately 60 members to ensure diversity and adequate representation, maintaining ≥50% participation in case of attrition. Panelists will complete a preliminary survey capturing professional background, experience, and geographic representation.

**Stage 4: Searching for evidence**

For each clinical question, a systematic review will be conducted and reported according to the PRISMA statement [13] and PRISMA-Search extension [14]. When quantitative synthesis is feasible, pairwise meta-analysis will be the preferred approach for studies addressing the same direct comparison and showing sufficient clinical and methodological comparability. Network meta-analysis will be considered only when the available evidence forms a connected network with at least one common comparator and when the assumptions of transitivity and overall comparability across studies are judged to be reasonable. If these conditions are not met, evidence will be synthesized through pairwise meta-analysis when appropriate or summarized narratively following the Synthesis Without Meta-analysis (SWiM) guideline [15]. If a network meta-analysis is undertaken, it will be reported according to the PRISMA-NMA extension [16].

For each clinical question, preliminary search strings may be drafted with the support of generative artificial intelligence, following the methodology outlined in "Using ChatGPT to Perform a Systematic Review: A Tutorial" [17], solely as an aid

for the initial formulation of search strategies. Final search strategies will be developed, reviewed, refined, and validated by the investigators, with methodological oversight from team members experienced in evidence synthesis and, when available, consultation with an information specialist or librarian. No AI-generated search strategy will be used directly without human verification. To ensure transparency and reproducibility, complete search strategies for each database will be provided as supplementary material in the final guideline. Literature searches will be conducted using conventional, reproducible search strategies in PubMed/MEDLINE, Embase, and the Cochrane Library, from database inception to December 2025, without restrictions on language or publication type. Expert opinion reviews, narrative reviews, case reports, and case series will be excluded.

If a systematic review addressing a given question has been published within the past five years, it will be considered as a primary source of evidence. However, an updated literature search will be conducted to identify newly published studies and ensure that the evidence base is current. When multiple systematic reviews addressing the same clinical question are identified, the most recent, comprehensive, and methodologically robust review will be prioritized. In cases of overlap or conflicting conclusions, an umbrella review approach will be undertaken, and discrepancies will be explored by examining differences in inclusion criteria, study selection, and risk-of-bias assessment. When necessary, primary studies will be re-evaluated to ensure consistency and completeness of the evidence synthesis.

Search results will be managed using Rayyan® [18], a validated web-based platform for systematic reviews. Duplicate records will be removed prior to screening. Titles and abstracts will be independently screened by two reviewers from the Steering Committee; full texts of potentially eligible studies will then be assessed independently, resolving discrepancies by consensus.

Data extraction will follow a standardized template, and study quality will be evaluated using AMSTAR-2 for systematic reviews, ROB 2.0 for randomized controlled trials, and ROBINS-I for non-randomized studies [19].

When quantitative synthesis is possible, random-effects models will be used to estimate pooled effect sizes and 95% confidence intervals. Heterogeneity will be assessed using the $I^2$ statistic, and publication bias will be evaluated through funnel plots and Egger's test.

## Stage 5: Formulating recommendations

The guideline will apply the GRADE (Grading of Recommendations, Assessment, Development and Evaluation) approach to ensure a transparent, systematic, and reproducible evaluation of the certainty of evidence and strength of recommendations.

The GRADE system classifies evidence certainty as high, moderate, low, or very low, based on study design, risk of bias, inconsistency, indirectness, imprecision, and publication bias. Recommendations will be categorized as strong or conditional (weak), depending on the balance between benefits and harms, evidence quality, stakeholder values and preferences, and resource considerations.

The GRADEpro GDT software (https://www.gradepro.org/) will be used to construct evidence profiles and Summary of Findings (SoF) tables for each key question.

## Stage 6: Reaching consensus on recommendations

Fig 3 outlines the Delphi process for recommendation consensus, including evidence-informed recommendation drafting, panel voting, feedback integration, and final approval.

Consensus on recommendations will be reached through a Delphi process, following the ACCORD guideline to ensure transparency and replicability. Based on the evidence summarized in the SoF tables and the corresponding GRADE ratings, the coordinators and steering committee will formulate recommendations for each question.

Participants will independently rate agreement using a five-point Likert scale, and two voting rounds will be conducted, incorporating feedback and statistical summaries after each round. The consensus threshold will be defined as ≥50% panel

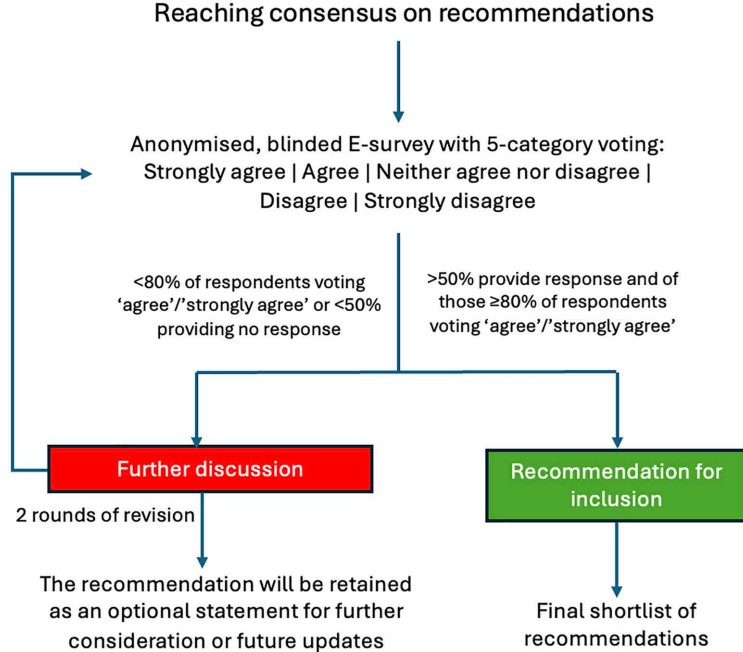

**Fig 3. Delphi consensus process for recommendations.**

participation and ≥80% agreement. The ≥50% participation threshold was selected to balance methodological rigor with the practical challenges of maintaining engagement across multiple Delphi rounds in a large and geographically diverse expert panel. Consensus will not be determined by participation alone but will also require a high agreement threshold of ≥80% among respondents. A participation rate of exactly 50% will be considered acceptable for that round. Reminder notifications will be sent during each Delphi round to optimize response rates and reduce attrition. If participation falls below 50%, the round will remain open for an additional reminder period before finalizing the results. Attrition between rounds will be documented, and response rates will be reported for each round. When feasible, respondent characteristics, including geographic region, practice setting, and professional background, will be summarized descriptively to assess the representativeness of the participating panel. Recommendations not reaching consensus after a Delphi round will be reviewed by the coordinators and steering committee together with the accompanying free-text comments. Recommendations may be modified between rounds when comments indicate ambiguity, redundancy, wording concerns, or the need for better alignment with the underlying evidence. Recommendations that remain below the predefined consensus threshold and are considered to have limited support may be dropped. Recommendations considered clinically relevant but not reaching consensus after two rounds will be retained as optional statements for further consideration or future updates.

Descriptive statistics—including response rates, agreement levels, medians, and interquartile ranges—will be used to evaluate consensus stability.

## Stage 7. Approval and dissemination

The final guideline will be discussed, reviewed, and approved during an in-person meeting of the GDG in Bogotá, Colombia. All participants will review and endorse the final version before submission. Those participating in all stages will be listed as co-authors, while partial contributors will be acknowledged (with consent).

The guideline will be published as an open-access document, ensuring broad accessibility.

To maximize dissemination, results will be presented at international surgical and academic congresses, and a comprehensive dissemination strategy will be implemented to promote the effective adoption of the guideline in clinical practice.

## Discussion

The procedure guide on laparoscopic cholecystectomy aims to establish a comprehensive and evidence-based framework for performing laparoscopic cholecystectomy according to the highest surgical standards. This guideline will provide a structured and transparent set of elements designed to optimize the technical and intraoperative execution of the procedure. Its development follows a systematic process, progressing from the formulation of exploratory clinical questions to highly structured, consensus-driven recommendations.

Although several existing guidelines and consensus-based initiatives have contributed substantially to improving the safety of laparoscopic cholecystectomy, including the prevention and management of bile duct injury, their scope is often centered on selected safety issues or specific technical steps rather than the full intraoperative workflow [20,21]. In contrast, the present procedure guide is designed to address a broader sequence of intraoperative domains, including access and setup, operative exposure, dissection strategies, use of adjunctive intraoperative techniques, documentation of safety steps, and bailout approaches in difficult scenarios. In addition, this project applies a structured methodology based on PICO-formulated questions, systematic evidence synthesis, Delphi consensus, and the GRADE approach to rate the certainty of evidence and strength of recommendations. These distinctions underscore the need for a globally applicable, evidence-based procedural guide that provides surgeons with clear, structured, and practical recommendations for performing laparoscopic cholecystectomy safely and effectively.

By systematically developing and disseminating this guideline, we seek to contribute to the global harmonization of laparoscopic cholecystectomy practice. The Procedure Guide aspires to become an international benchmark for surgical education, research, and clinical excellence, supporting both experienced surgeons and trainees in delivering safer, more consistent, and evidence-based care.

This protocol has several anticipated limitations. First, the evidence synthesis will rely primarily on published literature, which may not fully capture unpublished practices, local adaptations, or technical nuances that are not formally reported in the scientific literature. Second, although the Delphi panel is intended to include broad international and multidisciplinary representation, it may still not fully reflect practice variation across all regions, institutions, and healthcare environments. Third, the development of the guideline itself does not guarantee its implementation or clinical impact. The degree to which the final recommendations are adopted, accepted, and translated into improved outcomes will require subsequent evaluation through implementation studies, external validation, and future updates.

The Steering Committee welcomes inquiries and collaboration from surgeons, researchers, and professional societies interested in contributing to the development and dissemination of this initiative.

## Supporting information

**S1 Appendix. Questions.**
(DOCX)

## Acknowledgments

Andrés Isaza-Restrepo

Department of General Surgery, Hospital Universitario Mayor – Méderi, Bogotá, Colombia
ORCID iD: 0000-0002-1569-969X
andres.isaza@urosario.edu.co
Alberto Navarro-Alean

Department of General Surgery, Hospital Universitario Mayor – Méderi, Bogotá, Colombia

ORCID iD: 0000-0002-6382-6892

jorge.navarro@mederi.com.co

Susana Rojas-López

Department of General Surgery, Hospital Universitario Mayor – Méderi, Bogotá, Colombia

ORCID iD: 0009-0002-6599-3341

susana.rojas@mederi.com.co

Antonio Pesce

Department of Surgery, Azienda Unità Sanitaria Locale Ferrara, University of Ferrara, Via Valle Oppio, 2, 44023 Lagosanto, FE, Italy

ORCID: 0000-0002-7560-551X

José Alejandro Daza-Vergara

Research Department. Hospital Universitario Mayor – Méderi, Bogotá, Colombia

ORCID: 0000-0002-3810-3094

jose.daza@mederi.com.co

Luisa Fernanda Murcia-Soriano

Research Department. Hospital Universitario Mayor – Méderi, Bogotá, Colombia

ORCID: 0000-0003-2998-8561

luisa.murcia@mederi.com.co

Alejandro González-Muñoz

Hospital de Kennedy, Subred Sur Occidente E.S.E. Bogotá, Colombia

ORCID iD: 0000-0003-3890-3726

alegon_zalez@hotmail.com

Ana Carolina Buffara Blitzkow

Unit of General Surgery of Hospital de Clínicas da Universidade Federal do Paraná (UFPR) – Curitiba – Paraná – Brazil

ORCID iD: 0000-0003-1791-9892

anacarolina@mps.com.br

Arda Isik

Istanbul Medeniyet University, Istanbul

ORCID iD: 0000-0001-9493-4055

kararda@yahoo.com

Audrius Dulskas

National Cancer Institute, Vilnius, Lithuania and Vilnius University, Faculty of Medicine

ORCID iD: 0000-0003-3692-8962

audrius.dulskas@gmail.com

Camilo Andres Garcia Riaño

Hospital Internacional de Colombia – Fundación Universitaria FCV

ORCID iD: 0000-0001-5496-9413

camiloagarciar@gmail.com

Carlos Eduardo Rey Chaves

Pontificia Universidad Javeriana

ORCID iD: 0000-0001-6888-5595

carlosrey991@gmail.com

Danilo Osorio

Universidad del Cauca

ORCID iD: 0000-0002-1766-4722

danilof@unicauca.edu.co

Danny Conde-Monroy

LATAM AHPBA Hepatobiliary surgery fellowship

ORCID: 0000-0002-1365-4674

condedanny889@gmail.com

Diego Sierra Barbosa

Cirujano General – Profesor Universidad de La Sabana

ORCID iD: 0000-0002-0584-3897

diego.sierra@unisabana.edu.co

Ewen M Harrison

University of Edinburgh/ Royal Infirmary of Edinburgh

ORCID iD: 0000-0002-5018-3066

Ewen.Harrison@ed.ac.uk

Fabio Vergara Suárez

Cirujano hepatopancreatobiliar. Hospital internacional de Colombia (HIC)

ORCID: iD: 0000-0002-67382379

fabiovergara14@gmail.com

Fabrizio D'Acapito

U.O. Chirurgia Generale e Terapie Oncologiche Avanzate, Ospedale Morgagni-Pierantoni, Forlì, Italy

ORCID iD: 0000-0001-6420-6209

fabrizioda@gmail.com

Felipe Casas Jaramillo

Cirujano general La cardio

ORCID: 0000-0002-0174-519X

fcasasj@lacardio.org

Fernando Gutiérrez Infante

Especialista en entrenamiento en cirugía hepatopancreatobiliar Unisanitas

ORCID iD: 0000-0003-4412-9802

fernandogutierrezinfante@gmail.com

Gökhan Demiral

Recep Tayyip Erdoğan University, Faculty of Medicine, Department of General Surgery

ORCID iD: 0000-0003-2807-5437

drgokhandemiral@yahoo.com

Gustavo Martinez Mier, FACS

Department of Organ Transplantation, General Surgery & Division of Research. Unidad Médica de Alta Especialidad, Hospital de Especialidades No. 14, Centro Médico Nacional "Adolfo Ruiz Cortines", Instituto Mexicano del Seguro Social (IMSS). Veracruz 91897, Veracruz, Mexico

ORCID iD: 0000-0002-2883-9188

gmtzmier@hotmail.com

Ingrith Motta-Rincón

Departamento de Cirugía General – Hospital Naval de Cartagena – Armada Nacional de Colombia

ORCID iD: 0009-0008-8150-0408

imottarin@gmail.com

Ismael Domínguez Rosado

Departamento de Cirugía, Instituto Nacional de Ciencias Médicas y Nutrición Salvador Zubirán

ORCID iD: 0000-0002-5940-4208

Jorge David Peña Suárez

Clínica Reina Sofía, Clínicas Colsanitas

ORCID iD: 0000-0002-3516-2865

jorged.pena@urosario.edu.co

Jorge Muñoz Infante

Jefe de la Unidad de Educación. Centro Médico ISSEMYM

ORCID iD: 009-006-9167-8862

herniacentremexico@gmail.com

José Luis Quezada González

Filiación institucional: Hospital del Salvador/ Clínica Bupa/Universidad de Chile – Chile

ORCID iD: 0000-0003-0722-099X

jlquezadag@gmail.com

Juan Carlos Luna Cydejko

Clínica Internacional

ORCID iD: 0000-0002-0826-6589

consultas@doctorjcluna.com

Juan Pablo Muñoz Alzate

Anesthesiology, Universidad de Antioquia. Medellín, Colombia.

ORCID iD: 0009-0001-8975-2529

juanpabloma021191@gmail.com

Laura Covelli

Hospital Universitario Mayor – Méderi, Bogotá, Colombia

ORCID iD: 0000-0002-3329-9614

lau.x.covelli@gmail.com

Lovenish Bains

Department of Surgery, Maulana Azad Medical College & Lok Nayak Hospital, New Delhi- 110002, India

ORCID iD: 0000-0002-8627-0452

lovenishbains@gmail.com

Luis Gabriel González Higuera

Hospital universitario Nacional, Bogotá, Colombia

0009-0006-0155-4859

lugagonzalezh@hotmail.com

Marcello Di Martino

Department of Health Sciences, University of Piemonte Orientale, 28100 Novara, Italy.

ORCID iD: 0000-0001-6510-7210

marcello.dimartino@uniupo.it

Marcelo A. F. Ribeiro Jr.

University of Maryland – R Adams Cowley shock trauma center – Baltimore, MD, USA

ORCID iD: 0000-0001-9826-4722

mfribeiro@som.umaryland.edu

Marco Antonio Vanegas Cabrera

Hospital Universitario Mayor – Méderi, Bogotá, Colombia

ORCID iD: 0000-0002-5298-3825

marco9109@gmail.com

María Paula Moreno Knudsen

Hospital Universitario Mayor – Méderi

ORCID iD: 0009-0005-8262-0928

mpmoreknudsen@gmail.com

Mariana Ramírez Ceballos

Fundación Liga Ama Salvar Vidas – Pereira, Risaralda.

ORCID iD: 0000-0003-1171-5397

marianarceballos@gmail.com

Mohammed A. Omar

Qena University Hospitals, Qena University, Egypt

ORCID iD: 0000-0002-2736-8097

elqefty@yahoo.com

Moisés Barrientos Rivera

Hospitales La Paz (SERMESA), Guatemala, Guatemala

ORCID: 0009-0002-7302-4663

mbarrientos63@gmail.com

Mónica Gómez González

Department of General Surgery. Hospital Universitario Mayor – Méderi, Bogotá, Colombia

ORCID: 0000-0001-9788-4970

monicac.gomez@urosario.edu.co

Mónica Parrado Delgado

Cirujana General – Profesor Universidad de La Sabana

ORCID: 0009-0005-6666-1543

monipd2490@gmail.com

Natalia Andrea Rivera Rincón

Burjeel Hospital, Abu Dhabi – General Surgery Department

ORCID: 0000-0001-9801-6809

nataliaa.rivera@outlook.com

Néstor Vega Yuil

Hospital Santo Tomas/ Facultad de Médicina-Universidad de Panamá.

ORCID iD: 0009-0001-1555-1994

nestorvy@gmail.com

Oscar Rincón Barbosa

Hospital Militar

ORCID iD: 0000-0001-9605-4254

kelmvelx@gmail.com

Rafael Arraut-Gámez

Coordinador departamento de cirugía general – Hospital universitario Evaristo García HUV, sede Cartago – Valle del Cauca

ORCID iD: 0000-0002-0264-9853

reag14@hotmail.com

Raquel Tabares-Meza

Clínica Universitaria Colombia

ORCID iD: 0000-0002-3153-5695

raquel.tabares@gmail.com

Roberto Cirocchi

University of Perugia

ORCID iD: 0000-0002-2457-0636

roberto.cirocchi@unipg.it

Rocío Anula Fernández

Servicio de Cirugía. Hospital Clínico San Carlos, Instituto de Investigación Sanitaria San Carlos. Madrid, España

ORCID iD: 0000-0001-8112-1530

ranula@ucm.es

Saul Vargas-Rubiano

Hospital Universitario mayor – Méderi

saulvargas17@gmail.com

Sebastián Benavides Largo

Pontificia Universidad Javeriana, HUSI

ORCID iD: 0000-0002-3940-0162

sbenavides@husi.org.co

Sebastian Sierra Sierra

Universidad CES – Clinica CES, Medellín, Colombia

ORCID iD: 0000-0002-3253-028X

sebastiancirugia@gmail.com

Sergio Sanz

Universidad del Tolima – Hospital Federico Lleras

ORCID iD: 009-000-4996-6007

sersanz@yahoo.es

Shiva Jayaraman

St. Joseph's Health Centre – Unity Health Toronto; University of Toronto

ORCID iD: 0009-0004-4487-3521

Shiva.Jayaraman@unityhealth.to

Vicente E. Rodríguez-Maya

Hospital Clínica San Agustín, Loja – Ecuador

ORCID iD: 0000-0002-2692-0724

vicenterodma@gmail.com

Victor Manuel Quintero Riaza

Jefe seccion cirugia general, Hospital universitario Pablo Tobon Uribe Medellin

ORCID iD: 0000-0002-3889-6397

vquintero@hptu.org.co

Vishal Gupta

Department of Surgical Gastroenterology, All India Institute of Medical Sciences (AIIMS), Bhopal, MP 462020, INDIA

ORCID iD: 0000-0003-3574-1805

drvggis@gmail.com

Vishal G Shelat

Senior Consultant, Department of General Surgery, Tan Tock Seng Hospital, Singapore, 308433

ORCID iD: 0003-3988-8142

vgshelat@gmail.com

## Author contributions

**Conceptualization:** Camilo Ramírez-Giraldo, Alejandro Karduss-López, Daniela Álvarez-León.

**Methodology:** Camilo Ramírez-Giraldo, Alejandro Karduss-López, Daniela Álvarez-León.

**Project administration:** Camilo Ramírez-Giraldo, Alejandro Karduss-López, Daniela Álvarez-León.

**Supervision:** Alejandro Karduss-López, Daniela Álvarez-León.

**Visualization:** Camilo Ramírez-Giraldo.

**Writing – original draft:** Camilo Ramírez-Giraldo, Alejandro Karduss-López, Daniela Álvarez-León.

**Writing – review & editing:** Camilo Ramírez-Giraldo, Alejandro Karduss-López, Daniela Álvarez-León.

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
