## [Decision Letter · Decision Letter 0]

22 Feb 2026

PONE-D-26-02165Protocol for the Development of a Procedure Guide on Laparoscopic Cholecystectomy: Beyond Bile Duct Injury PreventionPLOS One

Dear Dr. Ramírez-Giraldo,

Thank you for submitting your manuscript to PLOS ONE. After careful consideration, we feel that it has merit but does not fully meet PLOS ONE’s publication criteria as it currently stands. Therefore, we invite you to submit a revised version of the manuscript that addresses the points raised during the review process.

A letter that responds to each point raised by the academic editor and reviewer(s). You should upload this letter as a separate file labeled ’Response to Reviewers’.A marked-up copy of your manuscript that highlights changes made to the original version. You should upload this as a separate file labeled ’Revised Manuscript with Track Changes’.An unmarked version of your revised paper without tracked changes. You should upload this as a separate file labeled ’Manuscript’.

We look forward to receiving your revised manuscript.

Kind regards,

Lovenish Bains, MS, FNB, FACS, FRCS (Glas), FICS, FIAGES

Academic Editor

PLOS One

Journal Requirements:

1. Please ensure that your manuscript meets PLOS ONE’s style requirements, including those for file naming. The PLOS ONE style templates can be found at

2. One of the noted authors is a group or consortium [Bogotá Task Force Collaboration Group]. In addition to naming the author group, please list the individual authors and affiliations within this group in the acknowledgments section of your manuscript. Please also indicate clearly a lead author for this group along with a contact email address.

Reviewers’ comments:

Reviewer’s Responses to Questions

**Comments to the Author**

1. Does the manuscript provide a valid rationale for the proposed study, with clearly identified and justified research questions?

Reviewer #1: Yes

Reviewer #2: Partly

Reviewer #3: Yes

Reviewer #4: Yes

Reviewer #5: Yes

Reviewer #6: Yes

2. Is the protocol technically sound and planned in a manner that will lead to a meaningful outcome and allow testing the stated hypotheses?

Reviewer #1: Yes

Reviewer #2: Partly

Reviewer #3: Yes

Reviewer #4: Yes

Reviewer #5: Yes

Reviewer #6: Partly

3. Is the methodology feasible and described in sufficient detail to allow the work to be replicable?

Reviewer #1: Yes

Reviewer #2: No

Reviewer #3: Yes

Reviewer #4: Yes

Reviewer #5: Yes

Reviewer #6: No

4. Have the authors described where all data underlying the findings will be made available when the study is complete?

Reviewer #1: Yes

Reviewer #2: No

Reviewer #3: Yes

Reviewer #4: Yes

Reviewer #5: Yes

Reviewer #6: Yes

5. Is the manuscript presented in an intelligible fashion and written in standard English?

Reviewer #1: Yes

Reviewer #2: Yes

Reviewer #3: Yes

Reviewer #4: Yes

Reviewer #5: Yes

Reviewer #6: Yes

6. Review Comments to the Author

You may also provide optional suggestions and comments to authors that they might find helpful in planning their study.

Reviewer #1: I would like to congratulate the authors on this well-designed protocol. The topic is highly clinically relevant. Although laparoscopic cholecystectomy remains the gold standard for the treatment of gallstone disease, the incidence of major bile duct and/or vasculobiliary injuries has not substantially decreased over the past two decades, despite the introduction and promotion of safety measures such as the Critical View of Safety (CVS) and broader initiatives including the Culture of Safety in Cholecystectomy (COSIC).

Given the very high volume of laparoscopic cholecystectomies performed worldwide, even an injury rate in the range of 0.4–0.6% translates into a significant burden for patients and healthcare systems. Severe biliary injuries are associated with considerable short- and long-term morbidity, the need for complex reparative procedures, impaired quality of life, and increased costs.

I hope that this work will contribute to standardizing intraoperative decision-making and surgical strategy, promoting a shared approach to difficult scenarios and, ultimately, improving patient safety.Overall, I believe the manuscript is suitable for acceptance in its current form. The only minor revision I would suggest is to reconsider the wording in the sentence “despite its routine nature, laparoscopic cholecystectomy is not without risk” (line 105). The term “routine” may inadvertently convey a sense of triviality, which contrasts with the manuscript’s focus on patient safety. A more neutral alternative could be: “Despite being frequently performed, laparoscopic cholecystectomy is not without risk.”

Reviewer #2: Main concerns and modification suggestions:

Use of ChatGPT in literature search strategy (key issue): On page 12, the authors mention, "Systematic literature searches will be performed with the assistance of ChatGPT 5.0." This is a serious methodological flaw and does not comply with the standards of systematic reviews. Using generative AI tools for systematic literature searches is unreliable and may lead to hallucinations (producing false literature), omissions, and reproducibility issues. PLOS ONE requires methods to be transparent and reproducible. This method must be removed and replaced with standard, reportable search strategies. Suggestion: Provide a detailed description of the search databases (such as PubMed/MEDLINE, Embase, Cochrane Library), search terms (including MeSH terms and free terms), search time frame, language restrictions, etc. It can be mentioned that professional medical librarians will be consulted to ensure the comprehensiveness of the search.

Specific operational details of the Delphi technique:

Consensus threshold: Setting "≥50% participation rate and ≥80% agreement rate" as the consensus standard is reasonable. However, it needs to be clarified how to handle cases where the participation rate is exactly on the 50% edge (for example, whether reminders will be sent after the first round to increase participation).

Composition and representativeness of the expert panel: Although a large number of experts (about 60) are listed, the criteria for expert selection (such as based on publication records, years of clinical experience, regional representation, etc.) need to be described in more detail to demonstrate their professionalism. Additionally, consideration should be given to including methodologists, patient representatives, or nursing representatives to enhance the comprehensiveness and applicability of the guidelines (AGREE II domain 3).

Role of "critical/important" classification: It is mentioned that experts will categorize issues as "critical," "important," or "not important." How this classification will affect subsequent processes (for example, whether only "critical" and "important" issues will be synthesized for evidence and recommendations) needs to be clarified.

Scope definition and clinical issues: It is clear that preoperative, postoperative management, and special populations are explicitly excluded. However, the scope of "intraoperative decision-making" is very broad. It is recommended that the authors provide 1-2 proposed PICO questions in the main text or supplementary materials to help readers better understand the specific concerns of the guidelines.

Evidence synthesis plan: The possibility of network meta-analysis (NMA) is mentioned. Given that comparative studies of laparoscopic cholecystectomy may lack directly comparable randomized controlled trials, the feasibility of conducting NMA needs to be assessed. The authors should clarify under what conditions (such as at least one common comparative group, study homogeneity, etc.) NMA will be considered; otherwise, it should be clearly stated that primarily paired meta-analysis or narrative summary will be conducted.

2. Presentation of results and data (for the protocol):

Main concerns and modification suggestions:

Data availability statement: The current statement "Deidentified research data will be made publicly available when the study is completed and published." is too vague. For a protocol, it should specify what types of data will be shared upon completion (for example, anonymized response data from the Delphi survey, extracted data tables from the systematic review, GRADE evidence profile tables, etc.), and the planned sharing platforms (such as OSF, Figshare, or as supplementary files to the article). This aligns with best practices in open science.

Ethics statement: Currently listed as "N/A." However, Delphi research involves collecting data and opinions from expert participants, which typically requires ethical review exemption or approval. The authors should confirm and supplement the corresponding ethical statement (for example, stating that the study has received approval or exemption from the ethics committee of XX institution, or that it does not require ethical approval because it falls under methodological research for guideline development, and clarify the reasons).

3. Discussion and conclusion:

Advantages: Clearly articulates the gaps this guideline aims to fill and its potential impact.

Main concerns and modification suggestions:

Critique of existing guidelines: Pages 13-14 mention that existing guidelines "lack methodological rigor." This assertion needs to be supported more cautiously and specifically. The cited references 20 and 21 (Agresta et al., 2015; Romero et al., 2021) are also consensus-based guidelines or Delphi studies. It could be pointed out that they have shortcomings in comprehensiveness or coverage of certain technical details, rather than broadly criticizing their methodological rigor unless there is clear assessment (such as using the AGREE II tool) to support this claim.

Limitations: As a protocol, the discussion section should include a discussion of the potential limitations of this protocol design. For example: reliance on published literature may not cover all unpublished surgical techniques; although the Delphi expert panel is international, it may still not represent practice differences across all regions; after the guidelines are developed, their actual adoption and implementation effects need to be validated by subsequent research.

Reviewer #3: This protocol addresses an important and well‑recognized gap in current surgical guidelines by proposing a comprehensive, evidence‑based approach to the intraoperative conduct of laparoscopic cholecystectomy. The methodology is generally rigorous, incorporating AGREE II, PRISMA, GRADE, and Delphi processes. The topic is clinically relevant and could have meaningful international impact.

However, substantial revisions are required to improve clarity, focus, and methodological transparency.

1. Move all of the contributors to a small table format and attach as supplementary material

2. Clarity on conflict management - describe how conflicts will be addressed, monitored, and managed.

3. Clarify how or if AI outputs will be validated by human information specialists

4. Standardizing ALL intra-op decisions seems unmanageable but possibly define a tighter scope of boundaries (ie. use of cholangiography, bailout methods like subtotal or fenestration, or different phases of the procedure)

5. 50% participation seems too low and should be more around 70% for validity

6. How will unresolved items be handled? what would criteria be to drop, modify or differ questions/recommendations?

7. Are there public or private stakeholders?

8. Provide rationale on size and composition of the expert panel

Reviewer #4: The authors plan to develop guidelines for safe laparoscopic cholecystectomy as most of the current guidelines focus only on bile duct injury prevention. The plan of developing guidelines is well formulated and it is in accordance with the current practices for development of similar guidelines

Reviewer #5: Though there is vast knowledge and guidelines available on the topic of the study, the authors are right in pointing out the lack of globally applicable and holistic guidelines that address the subject of laparoscopic cholecystectomy.

However, the proposed partcipants from membership of surgical society from a single country may not adequately address the topic for global consumption. At most, the study as proposed would serve as a national or regional guideline on the subject.

Further, the authors mention recruiting external experts via invitation by a steering committee. The criteria for inclusion of international experts is not well elaborated. The authors also mention ensuring broad representation ’across surgical specialties’ yet the topic is a guideline on a subspecialized area of general surgery. The aim should not be broad representation but subspecialized representation from the HPB/general surgeons with considerable experience on the subject.

I recommend that the authors tap into the memebrship of cross-border or international bodies such as IHBA or ILTS for the study to achieve the objectives and serve as global guideline or protocol on the subject.

Reviewer #6: Dear Author,

This manuscript presents a protocol for the development of an evidence-based procedural guideline for laparoscopic cholecystectomy, focusing on intraoperative technical conduct beyond bile duct injury prevention. The topic is clinically important and globally relevant, given the volume of procedures performed worldwide and ongoing efforts to standardize surgical safety practices.

The protocol demonstrates thoughtful planning and incorporates established methodological frameworks (AGREE II, PRISMA, GRADE, ACCORD, Delphi methodology). The international composition of the Guideline Development Group is a notable strength and enhances the credibility and potential global applicability of the initiative.

However, several areas require clarification to ensure methodological transparency, reproducibility, and clear differentiation from existing guidelines.

Major Comments

1. Clarification of Added Value Compared to Existing Guidelines

The manuscript states that current guidelines insufficiently address intraoperative technical conduct. While this may be true, the manuscript would benefit from a clearer and more structured comparison with major existing guidelines (e.g., WSES 2020 guidelines, multi-society safe cholecystectomy consensus, prior consensus-based recommendations).

To strengthen the justification for this initiative, the authors should:

• Explicitly outline how the proposed guideline differs in scope.

• Clarify whether it will cover a broader intraoperative workflow than existing initiatives.

• Specify how methodological rigor (e.g., systematic use of GRADE for all recommendations) distinguishes this effort.

A concise comparative table may improve clarity and strengthen the manuscript’s rationale.

2. Role of ChatGPT in Systematic Review Methodology

The manuscript indicates that systematic literature searches will be performed with assistance from ChatGPT 5.0. While innovative, this statement requires clarification to avoid concerns about reproducibility and methodological rigor.

The authors should clarify:

• Whether ChatGPT will assist only in drafting preliminary search strategies.

• Whether final search strategies will be validated by a trained information specialist or librarian.

• How transparency and reproducibility will be ensured.

• Whether complete search strategies will be provided as supplementary material.

Providing this clarification will strengthen confidence in the systematic review methodology.

3. Delphi Process and Participation Threshold

The defined consensus threshold (≥50% participation and ≥80% agreement) should be justified with reference to Delphi methodology standards. In particular, the rationale for a 50% participation threshold warrants explanation, as participation rates may influence representativeness.

The authors should also clarify how attrition between Delphi rounds will be handled and whether subgroup analyses (e.g., geographic representation, practice setting) are planned to ensure balanced input.

4. Clarification of Scope Boundaries

The guideline excludes preoperative and postoperative management, yet anesthesiologists are included for perioperative considerations. It would be helpful to clarify whether anesthesia-related aspects are strictly limited to intraoperative management.

Including one example PICO-formulated clinical question would enhance transparency and help readers understand the intended granularity and scope of recommendations.

5. Use of Existing Systematic Reviews

The manuscript proposes relying on systematic reviews published within the past five years when available. The authors should clarify:

• Whether updated searches will be conducted before adopting conclusions.

• How methodological quality will be assessed before incorporating existing reviews.

• How overlapping or conflicting reviews will be managed.

Clarifying these criteria will strengthen methodological consistency.

6. Data Availability Statement

As this is a protocol without primary data generation, the data availability statement should be clearly aligned with PLOS ONE’s policy for protocols and explicitly state that no datasets were generated or analyzed at this stage.

Minor Comments

1. “Paso 7” should be revised to “Stage 7.”

2. Minor typographical and spacing inconsistencies should be corrected.

3. Figures 1–3 would benefit from brief descriptive explanations in the main text.

4. The registration section should include the date of registration and confirm that it was prospectively registered.

This protocol addresses an important gap in the standardization of laparoscopic cholecystectomy technique. With clarification regarding AI-assisted search methodology, consensus thresholds, scope boundaries, and differentiation from existing guidelines, the manuscript would be substantially strengthened.

I encourage revision addressing the points above.

7. PLOS authors have the option to publish the peer review history of their article (what does this mean?). If published, this will include your full peer review and any attached files.

Reviewer #1: No

Reviewer #2: No

Reviewer #3: **Yes:**Matthew Factor

Reviewer #4: **Yes:**Sanjeev Kumar Gupta

Reviewer #5: No

Reviewer #6: **Yes:**DR MEGHA TANDON

---

## [Author Response · Author response to Decision Letter 1]

7 May 2026

Point-by-point response to reviewers´

Thank you for the valuable feedback and constructive reviewers’ comments aimed at improving this study.

Reviewer 1

1. Comment: Overall, I believe the manuscript is suitable for acceptance in its current form. The only minor revision I would suggest is to reconsider the wording in the sentence “despite its routine nature, laparoscopic cholecystectomy is not without risk” (line 105). The term “routine” may inadvertently convey a sense of triviality, which contrasts with the manuscript’s focus on patient safety. A more neutral alternative could be: “Despite being frequently performed, laparoscopic cholecystectomy is not without risk.”

Answer: We sincerely thank the reviewer for the kind and thoughtful comments, as well as for the positive overall assessment of our manuscript. We agree with this suggestion and have revised the sentence accordingly. The text now reads: “Despite being frequently performed, laparoscopic cholecystectomy is not without risk.”

Reviewer 2

1. Comment: Use of ChatGPT in literature search strategy (key issue): On page 12, the authors mention, "Systematic literature searches will be performed with the assistance of ChatGPT 5.0." This is a serious methodological flaw and does not comply with the standards of systematic reviews. Using generative AI tools for systematic literature searches is unreliable and may lead to hallucinations (producing false literature), omissions, and reproducibility issues. PLOS ONE requires methods to be transparent and reproducible. This method must be removed and replaced with standard, reportable search strategies. Suggestion: Provide a detailed description of the search databases (such as PubMed/MEDLINE, Embase, Cochrane Library), search terms (including MeSH terms and free terms), search time frame, language restrictions, etc. It can be mentioned that professional medical librarians will be consulted to ensure the comprehensiveness of the search.

Answer: We thank the reviewer for this important comment. We agree that the original wording may have suggested that generative AI would be used to conduct the systematic searches themselves, which was not our intention. We have therefore revised the manuscript to clarify that all final literature searches will be conducted using conventional, reproducible search strategies in PubMed/MEDLINE, Embase, and the Cochrane Library. For each clinical question, generative AI may be used only to assist in drafting preliminary search strings, which will then be critically reviewed, refined, and validated by the investigators before implementation. We have revised the Methods section accordingly to make the search process more transparent and reproducible.

2. Comment: Consensus threshold: Setting "≥50% participation rate and ≥80% agreement rate" as the consensus standard is reasonable. However, it needs to be clarified how to handle cases where the participation rate is exactly on the 50% edge (for example, whether reminders will be sent after the first round to increase participation).

Answer: We thank the reviewer for this helpful suggestion. We agree that the procedure for handling participation at the threshold should be clarified. We have revised the manuscript to specify that reminder notifications will be sent to participants during each Delphi round to optimize response rates. We also clarify that a participation rate of exactly 50% will be considered acceptable for that round, consistent with the predefined threshold of ≥50%. If participation falls below 50%, the round will remain open for an additional reminder period before finalizing the results. This clarification has been added to the Methods section.

3. Comment: Composition and representativeness of the expert panel: Although a large number of experts (about 60) are listed, the criteria for expert selection (such as based on publication records, years of clinical experience, regional representation, etc.) need to be described in more detail to demonstrate their professionalism. Additionally, consideration should be given to including methodologists, patient representatives, or nursing representatives to enhance the comprehensiveness and applicability of the guidelines (AGREE II domain 3).

Answer: We thank the reviewer for this thoughtful and important comment. We agree that the criteria for expert selection should be described in greater detail to better demonstrate the expertise and representativeness of the panel. We have therefore revised the manuscript to clarify that invited participants will be selected based on a combination of factors, including recognized clinical expertise in laparoscopic cholecystectomy or hepatobiliary surgery, academic and/or research experience, years of professional practice, and the aim of ensuring representation across different institutions, healthcare settings, surgical subspecialties, and geographic regions.

We also agree on the importance of multidisciplinary input. In response, we have made the role of methodological support more explicit in the manuscript and clarified that anesthesiologists were invited for questions specifically related to perioperative analgesia and anesthetic management. Because the present guideline is intentionally focused on intraoperative technical conduct, patient and nursing participation was not included in the initial Delphi panel; however, we acknowledge that these perspectives may be valuable for future implementation, dissemination, and updating phases of the guideline. We appreciate this suggestion and believe it has strengthened the transparency and applicability of the protocol.

4. Comment: Role of "critical/important" classification: It is mentioned that experts will categorize issues as "critical," "important," or "not important." How this classification will affect subsequent processes (for example, whether only "critical" and "important" issues will be synthesized for evidence and recommendations) needs to be clarified.

Answer: We thank the reviewer for this valuable comment. We agree that the role of the “critical,” “important,” and “not important” classification should be more clearly specified. We have revised the manuscript to clarify that this classification will be used as an additional prioritization tool during question selection and recommendation development. Questions rated predominantly as “critical” or “important,” and that also meet the predefined Delphi consensus threshold, will be prioritized for evidence synthesis and formulation of recommendations. Questions predominantly rated as “not important,” even if discussed during the Delphi process, will not be prioritized for full evidence synthesis unless the steering committee identifies a strong methodological or clinical reason for their retention. This clarification has been added to the Methods section.

5. Comment: Scope definition and clinical issues: It is clear that preoperative, postoperative management, and special populations are explicitly excluded. However, the scope of "intraoperative decision-making" is very broad. It is recommended that the authors provide 1–2 proposed PICO questions in the main text or supplementary materials to help readers better understand the specific concerns of the guidelines.

Answer: We thank the reviewer for this helpful suggestion. We agree that greater transparency regarding the proposed clinical questions would help readers better understand the scope of the guideline. In response, we have added the preliminary list of candidate clinical questions formulated by the Steering Committee as supplementary material and have cited this file in the manuscript. We believe this addition clarifies the specific intraoperative issues under consideration while preserving the readability of the main text.

6. Comment: Evidence synthesis plan: The possibility of network meta-analysis (NMA) is mentioned. Given that comparative studies of laparoscopic cholecystectomy may lack directly comparable randomized controlled trials, the feasibility of conducting NMA needs to be assessed. The authors should clarify under what conditions (such as at least one common comparative group, study homogeneity, etc.) NMA will be considered; otherwise, it should be clearly stated that primarily paired meta-analysis or narrative summary will be conducted.

Answer: We thank the reviewer for this important comment. We agree that the feasibility of network meta-analysis should be more clearly specified. We have revised the manuscript to clarify that pairwise meta-analysis will be the preferred quantitative synthesis approach whenever studies are sufficiently comparable and address the same direct comparison. Network meta-analysis will only be considered if the available evidence forms a connected network with at least one common comparator and if the assumptions of clinical and methodological comparability, as well as transitivity, are judged to be reasonable. If these conditions are not met, evidence will be summarized through pairwise meta-analysis when appropriate or by narrative synthesis following SWiM guidance. This clarification has been added to the Methods section.

7. Comment: Data availability statement: The current statement "Deidentified research data will be made publicly available when the study is completed and published." is too vague. For a protocol, it should specify what types of data will be shared upon completion (for example, anonymized response data from the Delphi survey, extracted data tables from the systematic review, GRADE evidence profile tables, etc.), and the planned sharing platforms (such as OSF, Figshare, or as supplementary files to the article). This aligns with best practices in open science.

Answer: We thank the reviewer for this helpful comment. We agree that the original data availability statement was too general for a protocol. We have revised the manuscript to specify the types of materials that will be shared upon completion of the study, including anonymized Delphi response data, extracted evidence tables from the systematic reviews, and GRADE evidence profile and Summary of Findings tables, when applicable. We also clarified that these materials will be made publicly available through the Open Science Framework and/or as supplementary files accompanying the final publication, in accordance with open science principles and journal requirements.

8. Comment: Ethics statement: Currently listed as "N/A." However, Delphi research involves collecting data and opinions from expert participants, which typically requires ethical review exemption or approval. The authors should confirm and supplement the corresponding ethical statement (for example, stating that the study has received approval or exemption from the ethics committee of XX institution, or that it does not require ethical approval because it falls under methodological research for guideline development, and clarify the reasons).

Answer: We thank the reviewer for this important observation. We agree that the previous ethics statement was insufficient and required clarification. The manuscript has been revised accordingly. This protocol was assessed by the corresponding institutional ethics committee and was approved through an expedited review process (approval code: CEISH-2026051; Minutes No. 040-2026). The study constitutes a low-risk methodological guideline-development project involving voluntary participation of expert panelists, with no patient enrollment, clinical intervention, or collection of sensitive personal data. In addition, Delphi responses were collected and analyzed in deidentified form.

9. Comment: Critique of existing guidelines: Pages 13–14 mention that existing guidelines "lack methodological rigor." This assertion needs to be supported more cautiously and specifically. The cited references 20 and 21 (Agresta et al., 2015; Romero et al., 2021) are also consensus-based guidelines or Delphi studies. It could be pointed out that they have shortcomings in comprehensiveness or coverage of certain technical details, rather than broadly criticizing their methodological rigor unless there is clear assessment (such as using the AGREE II tool) to support this claim.

Answer: We thank the reviewer for this thoughtful comment. We agree that our original wording was too broad and could be interpreted as an unsupported methodological judgment regarding previous consensus-based guidelines. Our intention was not to dismiss these efforts, but rather to highlight that existing documents do not comprehensively address the full range of intraoperative technical and decision-making aspects that this procedure guide aims to cover. We have therefore revised the Discussion section to adopt a more cautious and specific wording, focusing on limitations in scope, comprehensiveness, and coverage of technical details rather than making a general statement about methodological rigor.

10. Comment: Limitations: As a protocol, the discussion section should include a discussion of the potential limitations of this protocol design. For example: reliance on published literature may not cover all unpublished surgical techniques; although the Delphi expert panel is international, it may still not represent practice differences across all regions; after the guidelines are developed, their actual adoption and implementation effects need to be validated by subsequent research.

Answer: We thank the reviewer for this valuable suggestion. We agree that the Discussion section should acknowledge the potential limitations of the protocol design itself. In response, we have revised the manuscript to include a paragraph outlining key anticipated limitations, including the reliance on published evidence, the possibility that the expert panel may not fully capture all regional practice variations despite its international composition, and the need for future research to evaluate implementation, uptake, and real-world impact after the guideline is developed.

Reviewer 3

1. Comment: Move all of the contributors to a small table format and attach as supplementary material.

Answer: We thank the reviewer for this helpful suggestion. We agree that presenting the contributors in a table format as supplementary material improves the readability of the main manuscript. In response, we have removed the detailed contributor listing from the main text and included it in S2 Appendix.

2. Comment: Clarity on conflict management - describe how conflicts will be addressed, monitored, and managed.

Answer: We thank the reviewer for this important suggestion. We agree that the protocol should describe more explicitly how conflicts of interest will be addressed, monitored, and managed during the guideline development process. In response, we have revised the manuscript to clarify that all members of the guideline development process will be asked to disclose potential financial and non-financial conflicts of interest before participation. These disclosures will be reviewed by the coordinators and steering committee, documented, and considered when assigning roles in question formulation, evidence synthesis, and recommendation development. When deemed relevant, participants with potential conflicts may be restricted from voting or decision-making on specific questions or recommendations. This information has now been added to the Methods section.

3. Comment: Clarify how or if AI outputs will be validated by human information specialists.

Answer: We thank the reviewer for this important comment. We agree that human oversight of any AI-assisted output should be clearly stated. The manuscript has been revised to clarify that preliminary search strings drafted with the support of generative artificial intelligence will be reviewed, refined, and validated by the investigators before implementation in the databases, and that no AI-generated output will be used directly without human verification.

4. Comment: Standardizing ALL intra-op decisions seems unmanageable but possibly define a tighter scope of boundaries (ie. use of cholangiography, bailout methods like subtotal or fenestration, or different phases of the procedure).

Answer: We thank the reviewer for this thoughtful comment. We agree that the original wording may have suggested an overly broad or unmanageable

---

## [Decision Letter · Decision Letter 1]

14 May 2026

Protocol for the Development of a Procedure Guide on Laparoscopic Cholecystectomy: Beyond Bile Duct Injury Prevention

PONE-D-26-02165R1

Dear Dr. Ramírez-Giraldo,

We’re pleased to inform you that your manuscript has been judged scientifically suitable for publication and will be formally accepted for publication once it meets all outstanding technical requirements.

An invoice will be generated when your article is formally accepted. Please note, if your institution has a publishing partnership with PLOS and your article meets the relevant criteria, all or part of your publication costs will be covered. Please make sure your user information is up-to-date by logging into Editorial Manager at Editorial Manager® and clicking the ‘Update My Information’ link at the top of the page. For questions related to billing, please contact billing support.

Kind regards,

Wenguo Cui, Ph.D

Academic Editor

PLOS One

Additional Editor Comments (optional):

Reviewers’ comments:

Reviewer’s Responses to Questions

**Comments to the Author**

1. Does the manuscript provide a valid rationale for the proposed study, with clearly identified and justified research questions?

Reviewer #2: Yes

Reviewer #3: Yes

2. Is the protocol technically sound and planned in a manner that will lead to a meaningful outcome and allow testing the stated hypotheses?

Reviewer #2: Yes

Reviewer #3: Yes

3. Is the methodology feasible and described in sufficient detail to allow the work to be replicable?

Reviewer #2: Yes

Reviewer #3: Yes

4. Have the authors described where all data underlying the findings will be made available when the study is complete?

Reviewer #2: Yes

Reviewer #3: Yes

5. Is the manuscript presented in an intelligible fashion and written in standard English?

Reviewer #2: Yes

Reviewer #3: Yes

6. Review Comments to the Author

You may also provide optional suggestions and comments to authors that they might find helpful in planning their study.

Reviewer #2: The revised manuscript has fully addressed all the reviewers’ comments, the methodological flaws have been corrected, the ethical and data sharing statements have been improved, and the wording in the discussion section has become more precise. As a reviewer, I believe that the current quality of this manuscript’s revisions has reached the publication standards.

Reviewer #3: Based on my review of the revised manuscript, the authors have adequately addressed the reviewers’ comments, and I have no further concerns.

7. PLOS authors have the option to publish the peer review history of their article (what does this mean?). If published, this will include your full peer review and any attached files.

Reviewer #2: No

Reviewer #3: No

---

## [Editor Report · Acceptance letter]

PONE-D-26-02165R1

PLOS One

Dear Dr. Ramírez-Giraldo,

I’m pleased to inform you that your manuscript has been deemed suitable for publication in PLOS One. Congratulations! Your manuscript is now being handed over to our production team.

Lastly, if your institution or institutions have a press office, please let them know about your upcoming paper now to help maximize its impact. If they’ll be preparing press materials, please inform our press team within the next 48 hours. Your manuscript will remain under strict press embargo until 2 pm Eastern Time on the date of publication. For more information, please contact onepress@plos.org.

Kind regards,

on behalf of

Professor Wenguo Cui

Academic Editor

PLOS One